# Vesicular Messages from Dental Biofilms for Neutrophils

**DOI:** 10.3390/ijms25063314

**Published:** 2024-03-14

**Authors:** Ljubomir Vitkov, Jelena Krunić, Johanna Dudek, Madhusudhan Reddy Bobbili, Johannes Grillari, Bernhard Hausegger, Irena Mladenović, Nikola Stojanović, Wolf Dietrich Krautgartner, Hannah Oberthaler, Christine Schauer, Martin Herrmann, Jeeshan Singh, Bernd Minnich, Matthias Hannig

**Affiliations:** 1Department of Dental Pathology, Faculty of Medicine, University of East Sarajevo, 73300 Foča, Bosnia and Herzegovina; lvitkov@yahoo.com (L.V.); jelena.krunic@ues.rs.ba (J.K.); 2Clinic of Operative Dentistry, Periodontology and Preventive Dentistry, Saarland University, 664241 Homburg, Germany; johanna.dudek@uks.eu; 3Department of Environment & Biodiversity, University of Salzburg, 5020 Salzburg, Austria; bernhard.hausegger@gmail.com (B.H.); w.d.krautgartner@gmail.com (W.D.K.); bernd.minnich@plus.ac.at (B.M.); 4Ludwig Boltzmann Institute for Traumatology, Research Center in Cooperation with AUVA, 1200 Vienna, Austria; madhusudhan.bobbili@trauma.lbg.ac.at (M.R.B.); johannes.grillari@trauma.lbg.ac.at (J.G.); 5Department of Biotechnology, Institute of Molecular Biotechnology, University of Natural Resources and Life Sciences (BOKU University), 1190 Vienna, Austria; 6Department of Oral Rehabilitation, Faculty of Medicine, University of East Sarajevo, 73300 Foča, Bosnia and Herzegovina; irena.mladenovic@ues.rs.ba (I.M.);; 7Center for Biomedical Science, Faculty of Medicine, University of East Sarajevo, 73300 Foča, Bosnia and Herzegovina; 8Department of Internal Medicine 3—Rheumatology and Immunology, Universitätsklinikum Erlangen, Friedrich-Alexander-University Erlangen-Nürnberg (FAU), 91054 Erlangen, Germany; christine.schauer@uk-erlangen.de (C.S.); martin.herrmann@uk-erlangen.de (M.H.); jeeshan.singh@uk-erlangen.de (J.S.); 9Deutsches Zentrum für Immuntherapie (DZI), Universitätsklinikum Erlangen, Friedrich-Alexander-University Erlangen-Nürnberg (FAU), 91054 Erlangen, Germany

**Keywords:** periodontitis, dental biofilm, bacterial extracellular vesicles, outer membrane vesicles, caspase 4, trained immunity

## Abstract

The encounter between dental biofilm and neutrophils in periodontitis remains elusive, although it apparently plays a crucial role in the periodontal pathology and constitutes a key topic of periodontology. Dental biofilm and neutrophils were isolated from orally healthy persons and patients with periodontitis. We investigated biofilm and its particle-shedding phenomenon with electron microscopy and nanoparticle tracking analysis (NTA); biofilm shedding–neutrophil interactions were examined ex vivo with epi-fluorescence microscopy. For this purpose, we used acellular dental biofilm shedding, purified lipopolysaccharide (LPS), and phorbol 12-myristate 13-acetate (PMA) as activators, and the interleukin 8 receptor beta (CXCR2) inhibitor and the anti-interleukin 8 receptor alpha (CXCR1) antibody as modulators. The shedding of acellular dental biofilms overwhelmingly consists of bacterial extracellular vesicles (BEVs). The latter induced the moderate formation of neutrophil extracellular traps (NETs) in orally healthy subjects and a strong formation in patients with periodontitis. A CXCR2 inhibitor and an anti-CXCR1 antibody had a minor effect on NET formation. Neutrophils from patients with periodontitis exhibited NET hyper-responsiveness. BEVs were stronger inducers of NET formation than purified LPS and PMA. A plateau of neutrophil responsiveness is reached above the age of 40 years, indicating the abrupt switch of maladaptive trained immunity (TI) into the activated modus. Our results suggest that dental biofilms consist of and disseminate immense amounts of outer membrane vesicles (OMVs), which initiate NET formation via a non-canonical cytosolic LPS/caspase-4/11/Gasdermin D pathway. This modus of NET formation is independent of neutrophil elastase (NE), myeloperoxidase (MPO), peptidylarginine deiminase 4 (PAD4), and toll-like receptors (TLR). In periodontitis, the hyper-responsiveness of neutrophils to BEVs and the increased NET formation appear to be a consequence of TI.

## 1. Introduction

Periodontitis is a chronic gingival inflammation concomitant with alveolar bone resorption, resulting in teeth loosening and teeth loss in the long term. Etiologically, two categories are distinguished: (I) the rarely occurring early-onset periodontitis, predominantly characterised by neutrophil defects or neutropenia [1], and (II) late-onset periodontitis [2], characterised by neutrophil hyper-responsiveness [3] due to maladaptive trained immunity (TI) [4,5,6]. Neutrophils constitute the first line of defence in periodontal inflammation. The first encounter between neutrophils and dental biofilms occurs within the crevice, a space delineated by gingival and tooth surfaces. Dental biofilms are present in both periodontitis-prone and -resistant individuals. The neutrophils interact with biofilms (I) via soluble factors, such as toxins, proteases, pathogen-associated molecular patterns (PAMPs), and (II) via the endocytosis of disseminated biofilm bacteria. Neutrophil phagocytosis is a main defence mechanism, but crevicular neutrophils hardly phagocytose crevicular bacteria [7,8,9]. If and how the interactions of biofilms with neutrophils determine the neutrophil response in periodontitis is still unknown. Indeed, a crucial part of the host response in periodontitis is the formation of crevicular neutrophil extracellular traps (NETs) [10].

NETs are extracellular web-like fibres with a DNA backbone formed by activated neutrophils. They are largely composed of nuclear constituents that disarm and kill bacteria extracellularly, immobilize them, and thus prevent the colonization of new host surfaces [11]. The NET response may be triggered by the endocytosis of outer membrane vesicles (OMVs). The latter are spherical extracellular organelles of Gram-negative bacteria. The OMV cargo is enclosed by an outer membrane enriched with lipopolysaccharides (LPSs) [12], and they are frequently found as components of biofilms [13]. After endocytosis, OMVs are translocated from the early endosomal compartments into the neutrophils’ cytosols, where they activate the caspase-4/11/GSDMD-signalling pathway and NETs are formed [14,15,16,17]. OMVs have been reported in the in vitro-grown polymicrobial biofilms of *Porphyromonas gingivalis*, *Treponema denticola*, and *Tannerella forsythia* [18]. In general, biofilms are composed of bacteria, a fibrillary matrix, bacterial extracellular vesicles (BEVs) [19], and possibly some host-derived extracellular vesicles and vesicle-like structures that might descend from gingival crevicular fluid (GCF) [20]. The mechanical disruption of dental biofilm during mastication and daily oral hygiene results in the release of solitary bacterial cells into blood circulation [21,22]. Hence, we hypothesized that there may also be a concomitant release of multitudes of OMVs, as their release is an intrinsic feature of biofilm [23]. As dental biofilms are environmentally controlled [24], their effects on neutrophils can hardly be simulated by biofilms grown in vitro. Thus, even bodily fluids vastly alter the properties of biofilm BEVs, making them more pathogenic than BEVs from biofilms grown in vitro [25]. A further cardinal question arises: do neutrophils from patients with periodontitis react differently than those from orally healthy individuals?

The aim of this work was to examine (I) the difference between OMV-driven NET formation in patients with periodontitis and orally healthy subjects, and (II) to ascertain a conceivable relationship between the patients’ age and NET response.

## 2. Results

### 2.1. Dental Biofilms Are Rich in BEVs of Varying Morphologies

SEM examinations displayed plentiful subgingival biofilms on the roots of teeth extracted due to advanced periodontitis. Numerous vesicular structures were visible within the biofilm and on the bacteria of the dental biofilms (Figure 1A–D).

Our findings indicated that BEVs are regularly found in dental biofilm. TEM (Figure 1E,F) disclosed a multitude of BEVs and vesicle-like structures in the supernatant of a disrupted dental biofilm with largely varying dimensions, as previously reported for BEVs [24]. A few of these vesicles and vesicular structures might be of host origin [20]. The SEM pictures showed similar traits. Our results suggest that dental biofilms are rich in BEVs, which can be released in enormous amounts either spontaneously via biofilm dispersion [26] or the mechanical disruption of the biofilm [27], e.g., vortexing. Thus, released biofilm OMVs can deliver LPSs into the cytoplasm of neutrophils via endocytosis.

### 2.2. Quantification of BEVs

A nanoparticle tracking analysis (NTA) showed the hydrodynamic diameter (20–450 nm) of the vesicular structures, which were in agreement with the previously reported dimensions of a BEV, i.e., 20–400 nm [28]. The size distribution of both BEVs from orally healthy subjects and BEVs from patients with periodontitis varied within the range of 20–450 nm. Furthermore, we determined the median size (×50) of the BEVs between the two groups to be ranging from 127 to 183 nm (Figure 2A).

### 2.3. NET Morphology

The morphologic features are characteristic of NETs (Figure 3B insert) [29,30]. Importantly, all decondensed DNA was DNase I-sensitive. NETs induced by canonical stimulants proceed via a caspase-independent pathway and display the same morphology as the suicidal NET induced by caspase-4/5/11/gasdermin D (GSDMD), which is independent of peptidylarginine deiminase 4 (PAD4) [31]. Indeed, dental biofilm shedding barely affected citrullination in the neutrophils of orally healthy individuals (Figure 3A). In contrast, citrullination was more frequent in neutrophils from patients with periodontitis (Figure 3B).

Chromatin decondensation can be mediated by PAD4, which is activated by a spike in cytosolic calcium [32]. Since GSDMD pores enable a calcium influx [33], this suggests that GSDMD pores facilitate calcium-dependent PAD4 activation and the resultant histone citrullination [31]. Thus, the citrullination in neutrophils from patients with periodontitis might be a consequence of more intense NET formation, as PAD4 activation is a secondary process driven by GSDMD pores; H3 citrullination is time-dependent [34]. Thus, these results indicate quantitative differences between the patients and controls.

### 2.4. Site of NET Formation

As in the previous TEM studies of the pocket epithelium [35], we did not find NETs within the pocket epithelium or in the connective tissue of the gingiva. The meticulous TEM examination on the pocket’s surface revealed that only neutrophils in contact with GCF, i.e., exposed to OMV bombarding, formed NETs (Figure 4).

Adjacent cells, either epithelial ones or other neutrophils, efficiently protect the underlying epithelial cells against bombarding OMVs. Using ruthenium red/OsO4 staining enables the unambiguous differentiation between NETs and DNA protected by cell membranes [36]. Thus, the NET formation in periodontitis is initiated by the direct contact of neutrophils with GCF, i.e., with OMVs, which are unable to translocate through the adjacent host cells.

### 2.5. NET Quantification in Cultured Neutrophils

As the caspase-4/11-induced NET formation (Figure 4B) proceeds independently of myeloperoxidase (MPO), neutrophil elastase (NE), and PAD4 [31], we measured the NET via the quantification of cell-free DNA (cfDNA) in neutrophil cultures.

#### 2.5.1. cfDNA Quantification in Neutrophil Culture

The NET response in controls exhibited the highest value for BEVs, the response to PMA and LPS was slightly elevated. In patients with periodontitis, the NET responses to the culture medium, LPS, and dental biofilm shedding were elevated, and only the response to PMA remained similar to the controls. The response to dental biofilm shedding was considerably higher (Figure 5).

A *p*-value of ≤0.05 was considered significant.

In all patients with periodontitis, especially when stimulated with BEVs, NET responsiveness was enhanced compared to the healthy subjects (Figure 6A). The increased NET responsiveness in periodontitis correlates with neutrophil hyper-responsiveness [37,38].

#### 2.5.2. NET Dependence on Autocrine and Paracrine IL8

To specify the effects of the autocrine and paracrine interleukin-8 (IL8) on NET formation, we used both neutrophils without any inhibitors (Figure 5A) and with two selective CXCR inhibitors: (i) the CXCR2 inhibitor SB265610 (Figure 5B) and (ii) an anti-hCXCR1 antibody (Figure 5C). In orally healthy subjects, some inhibitory effects were observed in the PMA samples, but only insignificant ones were apparent in the biofilm supernatant samples. These findings indicate that the neutrophil autocrine and paracrine IL-8 insignificantly modulate NET biofilm supernatant-triggered formation, but their effects were rather small. Within patients with periodontitis, only the biofilm supernatant samples exhibited some irrelevant inhibitory effect on NET formation, indicating the minor role of autocrine and paracrine IL-8 in NET triggering in patients with periodontitis.

#### 2.5.3. NET Dependence on Age

NET formation by neutrophils from the control subjects increases with the participants’ age; however, NET responsiveness in the neutrophils of patients with periodontitis remains nearly unchanged, particularly relative to the biofilm supernatant (Figure 6).

The unchanged NET formation in the neutrophils of elderly patients with periodontitis seemingly contrasts with periodontal destruction, which increases in elderly orally healthy patients due to the accumulation of lifelong deterioration [39]. Age-unaffected NET formation in patients with periodontitis is compliant with the issue that NET hyper-responsiveness is a consequence of trained immunity [40,41].

### 2.6. Quantification of cfDNA in Blood Plasma

The plasma cfDNA is mostly of NET origin [42]. A comparison of the median values of cell cfDNA between the control and periodontitis groups shows no significant difference between the intersectional value of plasma cfDNA in the controls and the patients with periodontitis. However, the low number of participants in both groups limits the interpretation of these results.

## 3. Discussion

Currently, the biofilm is considered a resistant microbial form that is able to withstand the neutrophil challenge and most NETs [43,44]. However, the effects of dental biofilm OMVs on neutrophils remain elusive. Although studies on the in vitro-grown OMVs of planktonic periodontal pathogens have been reported previously [18], they have not examined the OMV properties of human dental biofilms. BEVs are environmentally controlled [24], and their pathogenicity vastly depends on biofilm host interactions [25]. Thus, using genuine human dental biofilms is essential for studying the clinically relevant effects of dental biofilm on neutrophils. Gram-negative bacteria predominate in subgingival biofilms. Thus, the main share of BEVs from dental biofilms in periodontitis comprises OMVs. They are released into the GCF by biofilm bacteria during normal cell growth without affecting bacterial viability, but growth conditions have a profound effect on the release of OMV [19,45]. Two main mechanisms are responsible for bacterial dissemination from biofilms: (I) bacterial dispersion, an active process controlled by intrinsic biofilm mechanisms, e.g., quorum sensing [26], and (II) detachment, a passive process driven by mechanical forces [27]. Subgingival dental biofilms are exposed during mastication and hygiene procedures to the pump-like pressure of the periodontal pocket. This is concomitant with bacterial translocation, a clear indication of biofilm detachment [21,22,46]. OMVs in crevices are highly loaded with LPS [47]; however, their clinical relevance has not yet been clarified.

The neutrophil response in periodontitis is characterized by extensive NET formation and a lack of phagocytosis [7,8,9]. NETs are restricted to the crevice, and there is the non-involvement of toll-like receptors (TLRs) [48] or CXCR1/2 in NET formation. These findings strongly indicate the non-canonical pathway of NET formation in periodontitis as a consequence of neutrophil–OMV interactions. Upon contact with the neutrophil cytoplasmic membrane, OMVs are endocytosed, and they release their LPSs from the early endosomal compartments into the neutrophils’ cytosol [47]. The cytosolic LPS induces the caspase-4/11/GSDMD-signalling pathway and subsequent NET formation [14,15,17,31] as caspase-4/5/11/GSDMD-driven NET formation is characterized by membrane permeabilization via GSDMD, and the lysis of phagocytosed bacteria cannot function. NETs are restricted to the crevice because it is rich in OMVs. As OMVs cannot penetrate the gingival tissue, no NETs were observed there.

Peripheral blood neutrophils from patients with periodontitis produced vastly more NETs when treated with the OMVs of the dental biofilm from patients with periodontitis than with the neutrophils from orally healthy individuals. This NET hyper-responsiveness to BEVs was restricted to patients with periodontitis, and this is due to maladaptive TI [40,41]. In periodontitis, the latter is driven by low-grade endotoxemia [4]. Dysregulated TI is prone to skew neutrophils into a non-resolving inflammatory state with elevated and reduced levels of inflammatory and homeostatic mediators, respectively [49]. In an animal model, the periodontitis-induced maladaptive TI phenotype and, consequently, the periodontitis can be transmitted to naive recipients via bone marrow transplantation [6].

At the age of 40 years, neutrophil hyper-responsiveness reached a plateau (Figure 6B) and did not depend on periodontal destruction. The latter is more pronounced in older patients as a consequence of the accumulation of lifelong deterioration rather than due to age-specific conditions [39]. The plateau of neutrophil hyper-responsiveness indicates the increasing transition of TI to an activated state. In contrast, a tendency to increase NET responsiveness was observed in control subjects, and this is likely a consequence of the accumulation of acquired minor immune defects.

The routine treatment approach in periodontitis consists of the reduction in dental biofilms. However, periodontal pathology comprises a chain of events: Bacterial biofilm-produced OMVs trigger an excess of NETs as a consequence of maladaptive trained immunity and the subsequent neutrophil hyper-responsiveness. As a result, this chain may be interrupted at any link; therefore, further studies on NET hyper-responsiveness to OMVs are needed. Moreover, the attenuation of TI-dependent neutrophil hyper-responsiveness might be a new treatment approach in periodontitis, as a decrease in neutrophil recruitment in experimental periodontitis attenuates bone resorption [50].

A limitation of the study is the relatively low number of participants due to the necessity to combine many experimental methods in several domains.

## 4. Materials and Methods

### 4.1. Patients’ Selection and Sample Collection

In all consecutively treated dental patients, orthopantomography, anamnestic questioning, and the clinical inspection of gingiva were routinely carried out. The inclusion criteria for patients with periodontitis were stage III and IV of periodontitis according to the current periodontitis classification [51,52,53]. The exclusion criteria were as follows: an age less than 35 years; necrotising periodontal diseases; abscesses of the periodontium; periodontitis associated with endodontic lesions; systemic diseases; antibiotic therapy in the last 6 months; treatment with steroids; and/or treatment with radiotherapy. The exclusion criteria for healthy controls were a lack of periodontal pathology; endodontic lesions; systemic diseases; antibiotic therapy in the last 6 months; treatment with steroids; and/or treatment with radiotherapy. All participants in this study provided informed written consent. The study protocol was approved by the ethics committee of the University of East Sarajevo, Faculty of Medicine Foča No. 01-2-40, and the study was conducted in accordance with the Helsinki Declaration as revised in 2013.

#### 4.1.1. Gingival Biopsies

Six patients (mean age 56 years) with advanced periodontitis and indicated teeth extractions were selected from consecutively treated dental patients for gingival biopsies. Radiological examination revealed an irregular bone resorption of more than 50%, and probing depth exceeded 6 mm. Immediately after the extraction of the tooth, a piece of pocket epithelium, with dimensions of nearly 2 mm × 2 mm, was excised using a scalpel and instantly washed twice in 50 mL of saline via swivelling for 20 s.

#### 4.1.2. Blood Sample Collection

Ten patients with periodontitis (six males and four females) aged between 45 and 64 years with a periodontal pocket depth of at least 6 mm were selected. Ten orally healthy subjects according to the current classification [51,52,53] (five males and five females) aged between 41 and 56 years were used as controls. Venous blood, in am amount of approximately 2.7 mL, was collected via venepuncture using a Safety-Multify^®^-Needle and Monovete^®^ EDTA K (Sarstedt, Nümbrecht, Germany).

#### 4.1.3. Dental Biofilm Collection

Dental biofilm was collected with a scaler at tooth necks from 10 patients with periodontitis. The collected dental biofilm was within the weight range of 4–15 mg from each patient. All samples were placed into separate Eppendorf tubes, stored at 4 °C, and processed within 60 min.

### 4.2. Electron Microscopy and Nanoparticle Tracking Analysis (NTA)

#### 4.2.1. Scanning Electron Microscopy (SEM)

Six teeth, extracted because of advanced periodontitis, were routinely fixed with Karnovsky fixative. The post-fixation of the samples was performed with 1% osmium tetroxide (buffered at pH 6.5 with 0.1 M of sodium cacodylate) for 2 h. Afterward, samples were transferred into a cacodylate buffer and postally sent to the University of Saarland, Germany, where they were dehydrated in an ascending series of ethyl alcohol, critical-point-dried, subsequently sputtered with gold (circa 5 nm), and examined in an environmental scanning electron microscope, ESEM XL30 (FEI Company, PHILIPS, Eindhoven, The Netherlands), operating at 20 kV.

#### 4.2.2. TEM

The gingival biopsies were fixed with 1.2% glutaraldehyde (buffered at pH 6.5 with 0.1 M of sodium cacodylate) with the addition of 0.05% ruthenium red for 2 h at room temperature, as applied for NET visualization [36], transferred into a cacodylate buffer, and postally send to the university of Salzburg, Austria, where post-fixation was performed with 1% osmium tetroxide (buffered at pH 6.5 with 0.1 M of sodium cacodylate) and 0.05% ruthenium red for 2 h at room temperature The specimens were routinely embedded in Epon 812, and ultrathin sections were cut and examined using a transmission electron microscope (LEO EM 910; LEO Elektronenmikroskopie Ltd., Oberkochen, Germany).

Biofilm supernatant processing: Nearly 1 mg of biofilm was supplemented with 200 µL of a cacodylate buffer. The samples in the cacodylate buffer were vortexed for 2 min and subsequently centrifuged at 6000× *g* for 10 min. Biofilm vortexing produces immense multitudes of BEVs with characteristic sizes [28]. The sediment was discarded, and the biofilm supernatant was pipetted off and filtered through a 0.2 µm bacterial filter. Only three drops per sample were dropped on parafilm and used for TEM; the remaining sample quantities were used for NTA, as described in Section 4.2.3. Formvar-coated copper grids were placed on the drops of the biofilm supernatant for one minute, blotted with filter paper, placed on a drop of Karnovsky fixative for another minute, blotted again, and then placed on two drops of bidistilled water and blotted. The grids were negatively stained via the use of 2% tungsten phosphate in bidistilled water after one minute of blotting and air-drying. The on-grid specimens were postally sent to the University of Saarland, Germany, and examined there using a transmission electron microscope, TECNAI 12 BioTwin (FEI Company, PHILIPS, Eindhoven, The Netherlands).

#### 4.2.3. NTA

The samples for NTA, as described in Section 4.2.2, were instantly fixed with a 4% paraformaldehyde and stored at −18 °C until shipping on dry ice to Ludwig Boltzmann Institute, Vienna, Austria. NTA was applied to access the particle concentration and size of all samples from control subjects and patients with periodontitis. All samples were characterized using a Zetaview PMX110 device from Particle Metrix (Zeta VIEW S/N 19-424, Software ZetaView 8.05.12 SP2, Camera 0.713 mum/px, Cell S/N: ZNTA, Particle Metrix, Meerbusch, Germany) in scatter mode. All samples were diluted to 1:100 in a mixture one-to-one of the supernatant buffer and the supernatant fixative. The diluted samples were filtered through a 0.22 µm filter and then measured in scatter mode with a 488 laser applying a shutter of 200 and a sensitivity of 85 to quantify particle concentrations and size. Each sample was measured in 3 technical replicates, and data were analysed using GraphPad Prism 9.0.0 (GraphPad Software, Inc., San Diego, CA, USA).

### 4.3. Neutrophil Isolation and Culture

Neutrophil separation was performed within two hours of venepuncture with the neutrophil isolation kit (EasySep™ Direct Human Neutrophil Isolation Kit, Stemcell Technologies, Vancouver, ON, Canada) and the appertaining magnet (EasySep™ Magnet, Stemcell Technologies, Vancouver, ON, Canada) according to the manufacturer’s guidelines. The neutrophil numbers were counted with Neubauer’s counting chamber. The purity of the PMN preparations was within the range of 95–96%. The neutrophils were incubated in Leibovitz’s L-15 medium supplemented with a 10% fetal bovine serum, 2 mg/mL of glucose, and a 1% Gibco™ antibiotic-antimycotic (100×) solution, catalog number 15240062. A number of 125,000 neutrophils per well were placed in 24-well plates and supplemented with either medium, 20 nM of PMA, or 10 µg/mL of free purified lipopolysaccharides from *E. coli* O26:B6, Sigma-Aldrich/Merck, Darmstadt, Germany, or 20 µL of biofilm supernatant, which was prepared in the following way: Nearly 1 mg of subgingival dental biofilm from each periodontitis patient was admixed and supplemented to 1.5 mL of Leibovitz’s L-15, vortexed for 2 min, and centrifuged at 2000× *g* for 5 min. The biofilm supernatant was collected and filtered through a 0.2 µm bacterial filter, and the sediment was discharged. Controls without inhibitors and two inhibitors were used: (i) SB265610 (Sigma-Aldrich/Merck, Germany) is a potent and selective IL8 receptor alpha (CXCR2) chemokine receptor antagonist that does not bind to IL8 receptor alpha (CXCR1), and (ii) the anti-hCXCR1 antibody, R&D Systems Inc., Minneapolis, MN, USA, binds human CXCR1 but does not cross-react with human CXCR2 and is routinely used as an inhibitor of CXCR1. The activated neutrophils were incubated at 37 °C for 3 h. Subsequently, NETs were digested with 1 U/mL of micrococcal nuclease, Thermo Fisher Scientific Inc., Waltham, MA, USA, for 10 min at 37 °C. The reaction was stopped with 20 mM of EDTA, the digested NET samples were shock-frozen, as well as the blood plasma samples, and both were shipped on dry ice to the University of Saarland, Germany, for DNA quantification.

### 4.4. Quantification of NETs from Cell Culture and Blood Plasma Cell-Free DNA

The Quant-iT™ High-Sensitivity DNA Assay kit (Thermo Fisher Scientific Inc., Waltham, MA, USA), was applied for cfDNA quantification. In total, 20 µL of each sample were prepared for fluorescence-based DNA quantification according to the manufacturer’s instructions. Fluorescence was detected using a Tecan Infinite M200 microplate reader (Tecan Trading AG, Männedorf, Switzerland). At the same time, DNA standards were measured. DNA quantities were determined through the resulting standard curves. Measuring the NET formation induced via caspase-4/5/11/GSDMD imposes a methodological study limitation, as caspase-4/11-induced NET formation proceeds independently of MPO, NE, and PAD4 [31] and, consequently, none of them may be used as a second quantitative parameter for caspase-4/5/11/GSDMD-driven NET quantification. For that reason, we used only cfDNA quantification and microscopy-acquired morphological features, including DNase I sensitivity, to confirm NETs.

### 4.5. Epifluorescence Microscopy

NETs were generated as described in Section 2.3 and adhered to slides placed in other 24-well plates, fixed with a 4% paraformaldehyde for 2 h, transferred into a cacodylate buffer, and postally sent to the university of Salzburg, Austria, where immune-fluorescent examinations were performed using anti-human antibodies for citrullinated histone H3 and the DNA molecular probe propidium iodide as previously reported [9]. The genuineness of NETs was confirmed via their digestion in control samples with 1 U/mL of micrococcal nuclease.

### 4.6. Statistics

Statistical analyses were performed using SigmaPlot 14.0 (Systat Software Inc., Point Richmond, CA, USA). Descriptive data analyses included means, standard deviations, standard errors, and C.I. of means, and inference statistics depending on the given data distribution (Shapiro–Wilk test) were used.

For statistical inference analysis, the following test procedures were applied: Fisher’s exact test, Kruskal–Wallis, a one-way analysis of variance on ranks, Student’s *t*-test, and the Mann–Whitney U rank sum test. The power (β) of the tests was set to 0.8, and the interval of confidence was set to 0.05 (α).

## 5. Conclusions

Dental biofilms disseminate immense amounts of BEVs. Endocytosed by neutrophils, OMVs deliver cytoplasmic LPS, which activates the non-canonical TLR-independent caspase-4/5/11/GSDMD-driven pathway of NET formation. The latter is stronger than that induced by free LPS, which is triggered by canonical MEK/ERK signalling. The neutrophils of patients with periodontitis are hyper-responsive to the OMVs of dental biofilms compared to neutrophils of orally healthy subjects. The NET hyper-responsiveness to the OMVs of dental biofilm in periodontitis appears to be the consequence of maladaptive TI. Attenuating the overreaction of neutrophils to form NETs may be a new treatment option for periodontitis.

## Figures and Tables

**Figure 1 ijms-25-03314-f001:**
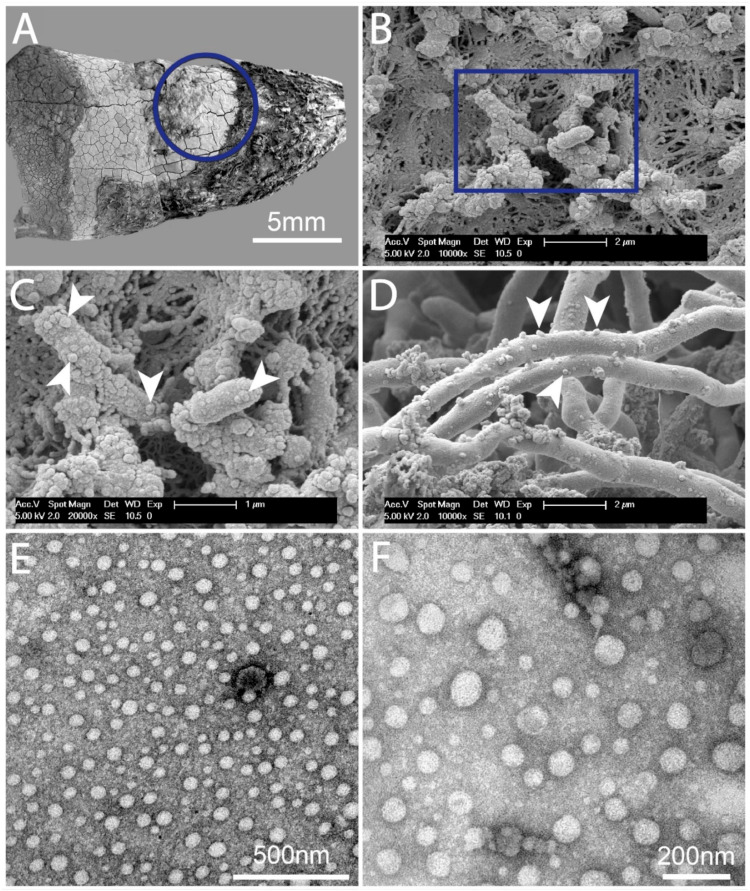
(**A**) The mesial root surface of an extracted human lower jaw molar with a huge buccal pocket. Top—buccal aspect; left—coronal aspect; bottom—lingual aspect; right—apical aspect. On the right, periodontium residual remnants are evident, the blue circle outlines the area of subgingival biofilms, where biofilm pictures were taken. On the left-most area, the dark grey is the supragingival biofilm. (**B**) An overview of the subgingival dental biofilm. Solitary bacteria and the fibrous biofilm matrix are evident. (**C**) A higher magnification of the marked area (blue square) in panel (**B**). Rod-like bacteria with multitudes of BEVs in between the fibrous biofilm matrix are evident. Arrowheads—BEVs. (**D**) Filamentous bacteria or hyphae with BEVs. Arrowheads—BEVs. (**E**) Overview of biofilm supernatant. (**F**) A higher magnification of the BEVs. The BEV dimensions shown are within the range of 20–450 nm. Structures smaller than 20 nm are protein aggregates [20].

**Figure 2 ijms-25-03314-f002:**
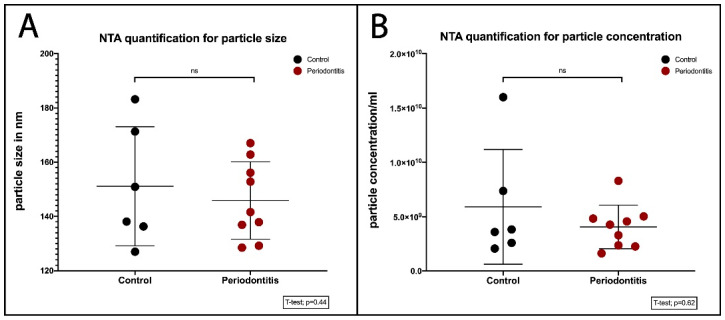
BEV size distributions are similar in controls and patients with periodontitis. (**A**) BEV size distribution. The dots represent the values of the BEV size and respective concentration distribution in the individual samples of the biofilm supernatant. Both parameters did not significantly (ns) differ between the controls and patients with periodontitis. (**B**) BEV concentration distribution. The concentrations of BEVs in the control and periodontitis groups ranged from 3 × 10^9^ to 1.6 × 10^10^ per mL. These results show that there is no significant difference in the concentration and size of BEVs derived from patients and controls (ns: *p* > 0.05) (Appendix A).

**Figure 3 ijms-25-03314-f003:**
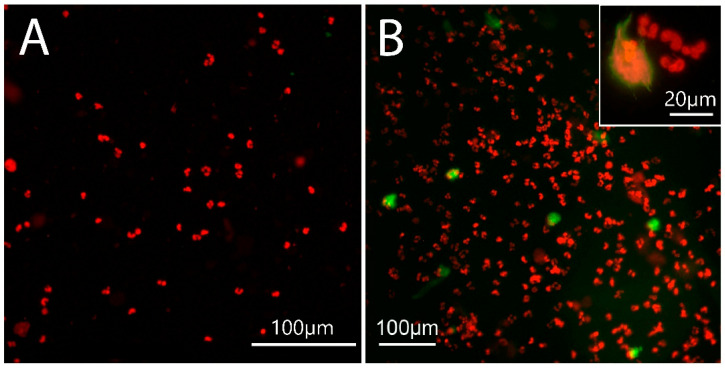
Epifluorescence microscopy. Citrullinated histone 3 (CitH3)—green; DNA—red. (**A**) Biofilm supernatant-stimulated neutrophils from a healthy oral volunteer. Only single neutrophils formed NETs, but they lacked citrullination. (**B**) Biofilm supernatant-stimulated neutrophils from a patient with chronic periodontitis. Many neutrophils form NETs and some of them are citrullinated (green). Insert: higher magnification of the NETs. NETs induced via caspase-4/5/11/GSDMD signalling [31] may share the same morphological features as canonical NETs.

**Figure 4 ijms-25-03314-f004:**
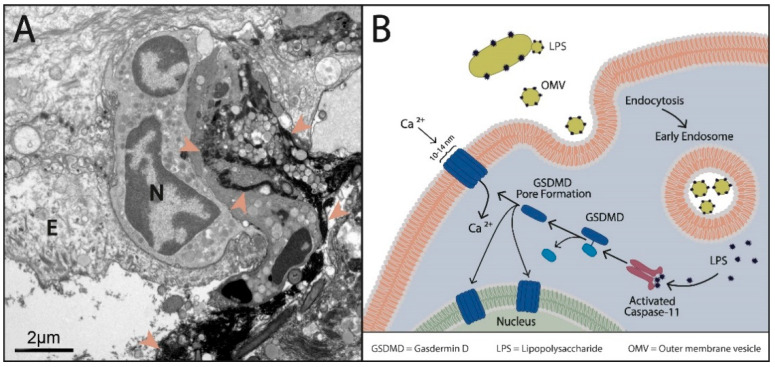
(**A**) Ruthenium red/OsO_4_ staining: E—epithelial cell; N—an intact neutrophil nucleus with and without contact to GCF; arrowheads—NETs derived from a neutrophil in contact with GCF. The extruded DNA is extensively blackened [36], as no cell membrane prevents the access of ruthenium red to DNA. (**B**) Schematic illustration of caspase 4/11-induced NETosis following OMV endocytosis.

**Figure 5 ijms-25-03314-f005:**
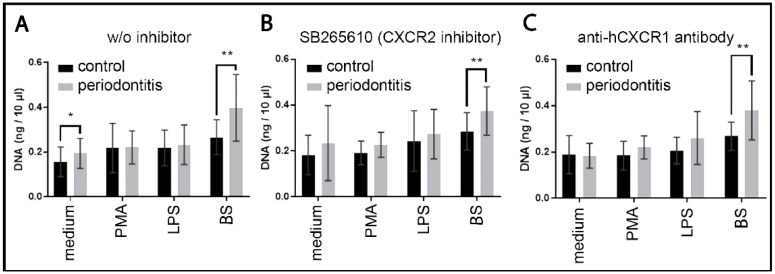
W/O inhibitor: without inhibitor; BS: biofilm supernatant; *: *p* ≤ 0.05; **: *p* ≤ 0.01. (**A**) Control without inhibitors. NET formation in controls and patients with periodontitis stimulated with the medium, PMA, LPS, and biofilm supernatant. (**B**) Effects of the interleukin 8 receptor beta (CXCR2) inhibitor and (**C**) effects of the anti-human interleukin 8 receptor alpha (CXCR1) antibody on NET formation (Appendix A).

**Figure 6 ijms-25-03314-f006:**
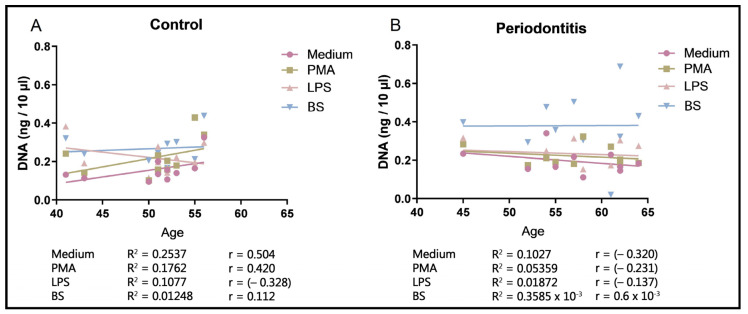
Age dependency of cfDNA in peripheral blood. (**A**) In orally healthy subjects, there is a tendency for a slight increase in NET responsiveness with age. (**B**) NET responsiveness in periodontitis, especially that of the biofilm supernatant, did not change with age. BS: biofilm supernatant (Appendix A).

## Data Availability

Data are contained within the article and Appendix A.

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
