# Peer review of "Vesicular Messages from Dental Biofilms for Neutrophils"

_ijms, 2024, doi:10.3390/ijms25063314_

Round 1

Reviewer 1 Report

Comments and Suggestions for Authors

Although this study is well-designed and focuses on the role of the neutrophil defensive mechanism in advanced periodontitis patients, I have some comments to make.

1. The title of this manuscript is too vague and, with regards to your study results, it needs to be revised with more specifics.

2. The introduction section needs to be updated to include the latest and relevant articles.

3. To enhance the research power of this study, it is necessary to add more samples due to the small sample size.

4. What is your ethical justification for collecting blood sample sizes from patients with periodontitis? Should have a deeper discussion about it.

5. Why not consider collecting saliva in this study and discussing it more?

6. It's important to compare your study results with relevant articles because the discussion part of this study is incomplete.

7. To increase the reader's motivation, it's important to include useful links between your results and clinical situations.

Comments on the Quality of English Language

Minor editing of English language required

Author Response

Although this study is well-designed and focuses on the role of the neutrophil defensive mechanism in advanced periodontitis patients, I have some comments to make.

  1. The title of this manuscript is too vague and, with regards to your study results, it needs to be revised with more specifics. "Changed according to the reviewer's suggestion"

  1. The introduction section needs to be updated to include the latest and relevant articles. "The basics of the NET triggering via the LPS-bearing OMVs are clearly stated in lines 74-82. The citations are dated from 2019 - 2021. This phenomenon has been not investigated in dentistry."

  1. To enhance the research power of this study, it is necessary to add more samples due to the small sample size. "Clinical studies on patients are effortful, costly and produce a lot of difficulties. Clinical studies with only four patients are also acceptable." The relatively low number of participants is denoted in “Results” as a study limitation.

  1. What is your ethical justification for collecting blood sample sizes from patients with periodontitis? Should have a deeper discussion about it. "For almost each investigation by internists or family physicians a routine blood count and other blood parameters are examined and usually larger blood samples are used. Patient consensus was taken and denoted (Line -), also Ethic Commission approval is in attached in the supporting file. Line 498-500: Institutional Review Board Statement: The study was conducted in accordance with the Declaration of Helsinki, and approved by the ethics committee of the University of East Sarajevo, Faculty of Medicine Foča N° 01-2-40 on 25.11.2021."

  1. Why not consider collecting saliva in this study and discussing it more? "Within the periodontal crevice there is no saliva. Crevicular neutrophils encounter saliva when they leave the periodontal crevice and this encounter does not affect the periodontitis."

  1. It's important to compare your study results with relevant articles because the discussion part of this study is incomplete.
    "No similar studies exist, the only compare was made with Hirsch", who used biofilm supernatant without knowing its vesicular nature and mechanism of action on neutrophils."

  1. To increase the reader's motivation, it's important to include useful links between your results and clinical situations.
    "added at the end of “Discussion””

The authors thank the reviewer for the helpful suggestions.

Reviewer 2 Report

Comments and Suggestions for Authors

Abstract

-It is not necessary to divide abstract into “Background, method, conclusion…..” parts.

-The author should briefly discuss the research aim.

Introduction

-NET?; as this term appears here for the first time.

-No explanation is given about periodontitis, the problems it causes, and the role of biofilm in the development of the disease.

- Why were patients less than 35 years old excluded from the study?

- What were the criteria for the diagnosis of periodontitis?

- Abbreviations should not be used in the captions of Figures.

- What software was used to draw the Fig.4B?

Line 200; Figure 6

-Line 124: Figure 2B

306: scanning 306 electron microscope; The full name is enough the first time.

Line 242; Figure 4B

Discussion; What suggestions do you have for future studies?

Comments on the Quality of English Language

Minor editing of English language required

Author Response

Abstract

-It is not necessary to divide abstract into “Background, method, conclusion…..” parts. "Changed according to the reviewer's suggestion"

-The author should briefly discuss the research aim. “Done according the reviewer's suggestion"

Introduction

-NET?; as this term appears here for the first time. “complemented, as suggested”

-No explanation is given about periodontitis, the problems it causes, and the role of biofilm in the development of the disease. “inserted at the begin of introduction”

- Why were patients less than 35 years old excluded from the study? “early-onset periodontitis is due to inborn neutrophil defects, begin in infancy and is extremely seldom, the late-onset periodontitis in age after 30 years” and affects 40% of the western world population. In order to find cases of advanced periodontitis (stage III and IV), we excluded patients below 35 years of age.”

- What were the criteria for the diagnosis of periodontitis? “They were inserted in 4.1. Patients’ selection

- Abbreviations should not be used in the captions of Figures. “corrected as suggested”

- What software was used to draw the Fig.4B? “added to the figure legend - Created with BioRender.com”

Line 200; Figure 6 “corrected”

-Line 124: Figure 2B “corrected”

306: scanning 306 electron microscope; The full name is enough the first time. “corrected”

Line 242; Figure 4B “removed”

Discussion; What suggestions do you have for future studies? “added at the end of “Discussion””.

The authors thank the reviewer for the helpful suggestions.

Reviewer 3 Report

Comments and Suggestions for Authors

Authors performed a study investigating the periodontal biofilm and its shedding with electron microscopy and NTA to investigate the Outer Membrane vesicles-driven NET formation in periodontitis patients vs a group oral orally healthy subjects. Several issues need to be solved. 

Please clarify your results in the abstract section: "The latter induced moderate and strong formation of neutrophil extracellular traps (NET) in orally healthy subjects and a strong one in patients with periodontitis, respectively." This sentence is not clear and it seems that no great differences exist between the two groups. 

I suggest to improve the introduction section focusing on the role of neutrophils and biofilm in the pathogenesis of periodontitis.

The aim of the study must be declared more clearly. 

I suggest to specify why this research could be relevant and could improve the actual knowledge of the argument. 

I suggest to revise the methods and results section. It is not clear how the selection of patients was made, and how you discriminate between pathological and non pathological patients. 

Comments on the Quality of English Language

Extensive editing of English language required

Author Response

Authors performed a study investigating the periodontal biofilm and its shedding with electron microscopy and NTA to investigate the Outer Membrane vesicles-driven NET formation in periodontitis patients vs a group oral orally healthy subjects. Several issues need to be solved.

Please clarify your results in the abstract section: "The latter induced moderate and strong formation of neutrophil extracellular traps (NET) in orally healthy subjects and a strong one in patients with periodontitis, respectively." This sentence is not clear and it seems that no great differences exist between the two groups. “rephrased as suggested”

I suggest to improve the introduction section focusing on the role of neutrophils and biofilm in the pathogenesis of periodontitis. “done as suggested”

The aim of the study must be declared more clearly. “complemented as suggested”

I suggest to specify why this research could be relevant and could improve the actual knowledge of the argument. "added at the end of “Discussion””

I suggest to revise the methods and results section. It is not clear how the selection of patients was made, and how you discriminate between pathological and non pathological patients. “Inclusion criteria were added, they contain the diagnostic criteria”

The authors thank the reviewer for the helpful suggestions.

Round 2

Reviewer 1 Report

Comments and Suggestions for Authors

All my concerns were partially addressed and I have no additional comments.

Comments on the Quality of English Language

Minor editing of English language required

Author Response

Thank you!

Reviewer 2 Report

Comments and Suggestions for Authors

Dear Editor, the authors have answered the questions, but I have some comments and recommendations:

Introduction part

Line 54: in the long term NOT “the in long term”

The criteria for the diagnosis of periodontitis should be entered in the method section. periodontal parameters, microbial parameters, or, ..?

The reasons for excluding people under 35 years of age should be given in the text

fig-àFig “throughout the text”

First author cited more than 5 times.

Comments on the Quality of English Language

Minor editing of English language required

Author Response

The reasons for excluding people under 35 years of age should be given in the text. “They are given - advanced periodontitis with stage III and IV of late-onset periodontitis can be hardly found in age under 35years – this is a fact, well known to each dentist.”

fig-àFig “throughout the text” "We changed fig. into figure".

First author cited more than 5 times. "Self-citation are undesirable in reviews, but this is an original research. Concerning NETs in periodontitis Vitkov is ever cited, as his works and those of Hirschfeld are the only ones using biopsies and dental biofilms from humans."

Minor editing of English language required "English editing was performed"

Reviewer 3 Report

Comments and Suggestions for Authors

The authors respond to almost all my queries but I suggest to modify the sentence related to the aim of the work. "The aim of this work was to (I) obtain human dental biofilms (II) isolate and quantify  OMVs in a clinically relevant setting (III) examine their structure, and (IV) investigate  OMV-driven NET formation in patients with periodontitis and orally healthy subjects,  and (V) ascertain a conceivable relationship between the patients’ age and NET response."In this sentence only the 5th point could be considered an aim of this study.

Comments on the Quality of English Language

Moderate editing of English language required

Author Response

The authors respond to almost all my queries but I suggest to modify the sentence related to the aim of the work. "The aim of this work was to (I) obtain human dental biofilms (II) isolate and quantify  OMVs in a clinically relevant setting (III) examine their structure, and (IV) investigate  OMV-driven NET formation in patients with periodontitis and orally healthy subjects,  and (V) ascertain a conceivable relationship between the patients’ age and NET response."In this sentence only the 5th point could be considered an aim of this study. "This sentence was modified."

Moderate editing of English language required “English editing was performed”